# Understanding the AIE phenomenon of nonconjugated rhodamine derivatives via aggregation-induced molecular conformation change

Lin-Lin Yang ®[1,2,5], Haoran Wang ®[2,3,4,5], Jianyu Zhang ®[3], Bo Wu[2], Qiyao Li[2], Jie-Ying Chen[1], A-Ling Tang[1], Jacky W. Y. Lam ®[3], Zheng Zhao[2,4] ✉, Song Yang ®[1] ✉ & Ben Zhong Tang ®[2,3,4] ✉

The bottom-up molecular science research paradigm has greatly propelled the advancement of materials science. However, some organic molecules can exhibit markedly different properties upon aggregation. Understanding the emergence of these properties and structure-property relationship has become a new research hotspot. In this work, by taking the unique closed-form rhodamines-based aggregation-induced emission (AIE) system as model compounds, we investigated their luminescent properties and the underlying mechanism deeply from a top-down viewpoint. Interestingly, the closed-form rhodamine-based AIE system did not display the expected emission behavior under high-viscosity or low-temperature conditions. Alternatively, we finally found that the molecular conformation change upon aggregation induced intramolecular charge transfer emission and played a significant role for the AIE phenomenon of these closed-form rhodamine derivatives. The application of these closed-form rhodamine-based AIE probe in food spoilage detection was also explored.

Unveiling the structure-property relationship of organic functional materials and proposing effective materials design principle are among the most fascinating tasks for material scientists[1–3]. Current molecular design of functional materials mainly follows a bottom-up molecular science research paradigm, which emphasizes the role of molecular structure in determining the properties of materials[4–6]. Under the guidance of this molecular structure dominated research paradigm, various molecular materials with new structures have been designed through an experience-based subconsciousness. Taking organic luminescent materials as an example, the introduction of conjugated rigid building blocks and the expansion of the π-conjugation are the commonly used strategies to develop new luminescent materials since they are generally thought necessary for the formation of chromophore and tuning the band gap[7–9]. Another example is the preparation of chiral materials or drugs, which usually needs the incorporation of chiral factors or asymmetry catalysis to induce the generation of a chiral center[10].

[1]National Key Laboratory of Green Pesticide, Key Laboratory of Green Pesticide and Agricultural Bioengineering, Ministry of Education, Guizhou University, Huaxi District, Guiyang 550025, China. [2]Clinical Translational Research Center of Aggregation-Induced Emission, The Second Affiliated Hospital, School of Science and Engineering, Shenzhen Institute of Aggregate Science and Technology, The Chinese University of Hong Kong, Shenzhen (CUHK-Shenzhen), Guangdong 518172, P.R. China. [3]Hong Kong Branch of Chinese National Engineering Research Center for Tissue Restoration and Reconstruction, Department of Chemistry, The Hong Kong University of Science and Technology, Clear Water Bay, Kowloon, Hong Kong 999077, China. [4]HKUST Shenzhen Research Institute, No. 9 Yuexing 1st RD, South Area Hi-tech Park Nanshan Shenzhen 518057, China. [5]These authors contributed equally: Lin-Lin Yang, Haoran Wang. ✉e-mail: zhaozheng@cuhk.edu.cn; jhzx.msm@gmail.com; tangbenz@cuhk.edu.cn

It is no doubt that the molecular structure dominated research paradigm has made significant contributions to the advancement of modern chemistry and materials[11]. However, this molecular structure oriented materials development approach also suffers some emerging limitations with the expansion of the materials scope. There are more and more cases indicated that the molecular structure alone cannot fully determine the materials properties while their aggregation modes greatly influence their macroscopic performance[12,13]. For instance, Yuan and coworkers have discovered that the chromophore-free saccharide molecules without even one phenyl ring could exhibit strong luminescence upon aggregation, and polypeptide and some other non-conjugated polymers exhibited similar phenomenon[14,15]. Tang and coworkers discovered another type interesting luminescence phenomenon that some propeller-shaped molecules exhibit no emission in their isolated state but show bright emission in aggregation state, which they has coined as aggregation-induced emission (AIE)[16,17]. Li and Zhang et al. reported that the achiral molecules like hexaphenylsilole and tetraphenylethene (TPE) derivatives with non-chiral centers could generate circular dichroism signals and even circularly polarized light emission in aggregates while their single molecular state didn't show any chiral signals[18–20]. Furthermore, in the field of biology, the folded 3D structure of polypeptide and its wrongly folded structure-resulted aggregates could also exhibit vastly different biological activities[21,22]. These inconsistences between molecular properties and aggregate properties can possibly be ascribed to the complex changes occurring at both molecular level and aggregate level during the aggregation process, which unfortunately has long been ignored as the role of molecular structure is overpriced. Therefore, it is highly desirable to revisit the structure-property relationship from a top-down perspective with more attention focus on the influence brought by aggregation to the molecules.

By shifting the research interest from single molecule to aggregates at condensed state, scientists have unveiled that lots of factors induced by the multibody interactions play an important role in determining the materials performance[23–25]. For example, An at al. reported that H-aggregation in molecular packing could stabilize triplet excitons to realize ultralong organic phosphorescence[26]. Li et al. established the one-to-one correspondence between the determinate interactions and excited triplet states, which guided the design of

organic phosphorescence materials[27]. Barrett at al. reported that co-crystallization of two or more molecules can exhibit properties that do not belong to any individual molecules[28–30]. These works also highlighted how did the crystallinity, aggregate morphology, assembly behavior that occurred among multibodies affect the materials performance[31,32]. In addition to multibody interactions, whether the aggregation will cause the change of the characteristic of molecule itself, like the molecular conformation or molecular electronic structure, leading to emerging of new macroscopic materials performance is also an interesting topic that is worth to investigated. However, investigation from this top-down viewpoint has seldomly been considered and there are very few related studies. Thus, it is crucial to investigate how the spatial confinement resulting from aggregation and changes in the surrounding molecular environment affect molecular characteristic like the molecular conformation. In other words, further research is needed to understand the differences in molecular conformation between the single molecular state and the aggregate state. This investigation will provide valuable insights into the influence of aggregation on materials properties and enable the development of strategies to optimize their performance.

Open-form rhodamine compounds are a kind of widely used luminescent dyes with a planar π-conjugated xanthene core and high absorption coefficient, high fluorescence quantum yield, water solubility and biocompatibility. While the closed-form rhodamines generally are non-emissive due to the lack of conjugated chromophore. Therefore, the closed-form rhodamine derivatives usually need to transform to open-form under external stimuli to emit light, which could be utilized to design various turn-on fluorescent probes and biosensors[33]. To further fulfill the application requirements in the aggregate state, the open-form rhodamine compounds need to be further modified with TPE or other bulky and twisted building blocks (Fig. 1a) according to the AIE mechanism of restriction of intramolecular motion (RIM)[34–36], which is rather complicated in design and challenge in synthesis. In this work, a series of AIE-active closed-form rhodamine derivatives (Fig. 1b) are reported with strong aggregate emission in the visible region although no identifiable chromophores are contained. Further mechanism study showed that the closed-form rhodamine-based AIE system did not strictly follow the classic RIM mechanism which has been widely used to explain the AIE

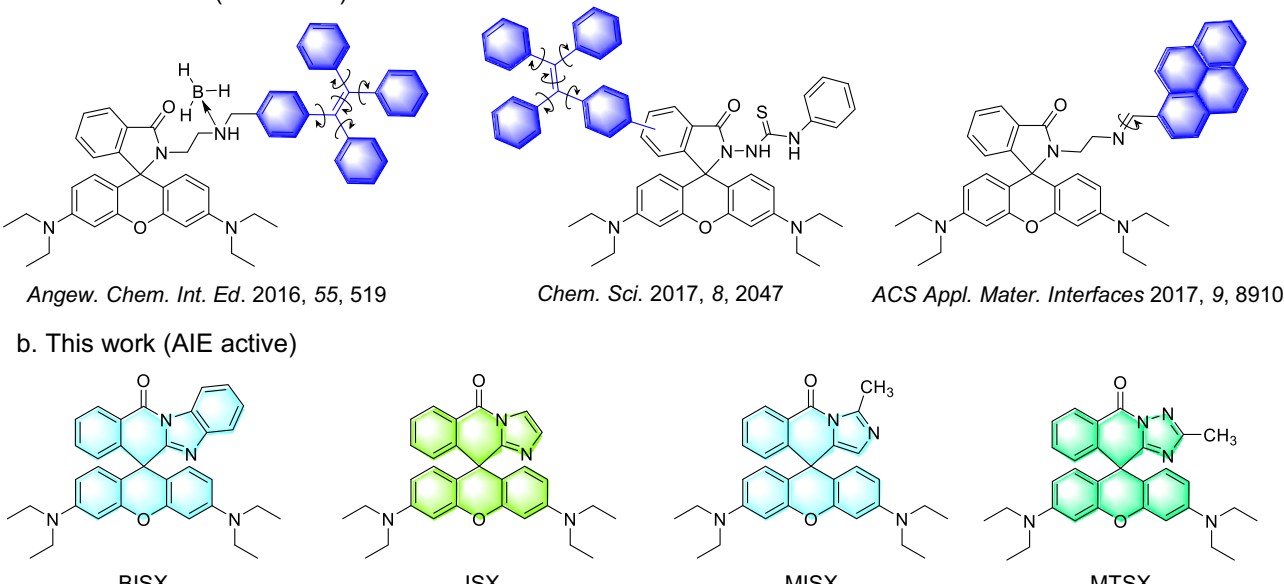

**Fig. 1 | Chemical structures of the Rhodamine B-based AIEgens. a** Previously reported work on Rhodamine B-based AIEgens via introducing rotors. **b** Unique rotor-free Rhodamine B-based AIEgens (BISX, ISX, MISX and MTSX) reported in this work. The filled colors indicate their crystal emission, respectively.

phenomenon. Through a top-down viewpoint, we finally found that the molecular conformation change upon aggregation induced intramolecular charge transfer emission and played a significant role in the AIE phenomenon of these closed-form rhodamine derivatives. Food spoilage detection of seafood could be realized with the closed-form rhodamine-based AIE probe, showing their promising prospect for portable prototypes.

## Results

### Structures and photophysical properties

The target compounds so-named BISX, ISX, MISX and MTSX were readily achieved via a facile synthetic route from their corresponding imidazole or triazole derivatives using the C−N and C−C coupling cyclization reaction with rhodamine B chloride, according to our previous work[37]. Details of the experimental procedures are provided in the Supplementary Information (Supplementary Fig. 1). DFT calculations were carried out to investigate the reaction mechanism of MISX (Supplementary Fig. 2). All compounds were fully characterized by $^1$H NMR and $^{13}$C NMR spectroscopies, high-resolution mass spectrometry, and the obtained single-crystal structures, respectively (Supplementary Figs. 3–14, Supplementary Table 1).

After confirming the structure, we investigated their UV−vis absorption and photoluminescence (PL). As shown in Supplementary Fig. 15, the four compounds displayed similar characteristic absorption spectra with several sharp bands between 240 nm to 330 nm in THF solutions, which could be attributed to their intramolecular n-π* transition and π-π* transition. The molar absorption coefficient of BISX, ISX, MISX and MTSX is measured as 7446 M$^{-1}$·cm$^{-1}$, 7146 M$^{-1}$·cm$^{-1}$, 7700 M$^{-1}$·cm$^{-1}$, and 7566 M$^{-1}$·cm$^{-1}$, respectively. To explore their AIE feature, PL spectra were measured in THF/water mixtures, respectively (Fig. 2a−d). For BISX, when the water fraction was below 80%, it was non-emissive. However, when the water fraction reached 90%, an intensified emission peak at 510 nm emerged. And the emission amplified at $f_w = 99\%$ with a 25-fold increase relative to that in THF

solution, displaying a bright green fluorescence (Fig. 2e). The other three compounds (ISX, MISX and MTSX) showed a similar pattern, with their emission intensity in the aggregate state enhanced to about 35, 80, and 35-fold compared to that of the THF solution, respectively (Fig. 2f−h). The aggregate formation was manifested by the level-off tails in the long-wavelength region of the absorption spectra in the aqueous mixtures due to the Mie effect of the dye nanoparticles (Supplementary Fig. 16). Therefore, all four compounds presented typical AIE features, with aggregates formation confirmed by dynamic light scattering analysis (Supplementary Fig. 17). For all of the four compounds, their lifetimes in the crystal state are one order of magnitude higher than that in the solution state, which matched their enhanced emission from solution to solid state (Supplementary Figs. 18–21).

### Mechanism of AIE study

To disclose the AIE feature of the four compounds, firstly, we measured their emission in the methanol/glycerol mixtures with varied glycerol fractions, since typical AIE molecules generally show enhanced emission in high viscous media due to the RIM mechanism. To our surprise, for all four compounds, increasing viscosity of the mixture had no impact on the fluorescence intensity (Fig. 3a and Supplementary Fig. 22). Then a classical cooling experiment was performed. As shown in Fig. 3b, c, when gradually decreasing the temperature of BISX dissolved in THF from ambient temperature to −196 °C, its AIE emission peak at 540 nm barely enhanced. Instead, an emission peak at a short wavelength enhanced gradually, and so did the other three compounds (Supplementary Fig. 23). These unexpected phenomena indicated that these AIEgens may not be determined by the conventional RIM working mechanism[38,39], further arousing our attention.

As these cross-shaped molecules are separated by two moieties, the xanthene part and isoquinolinone part, elucidating which part serves as the chromophore became significant to understand the AIE

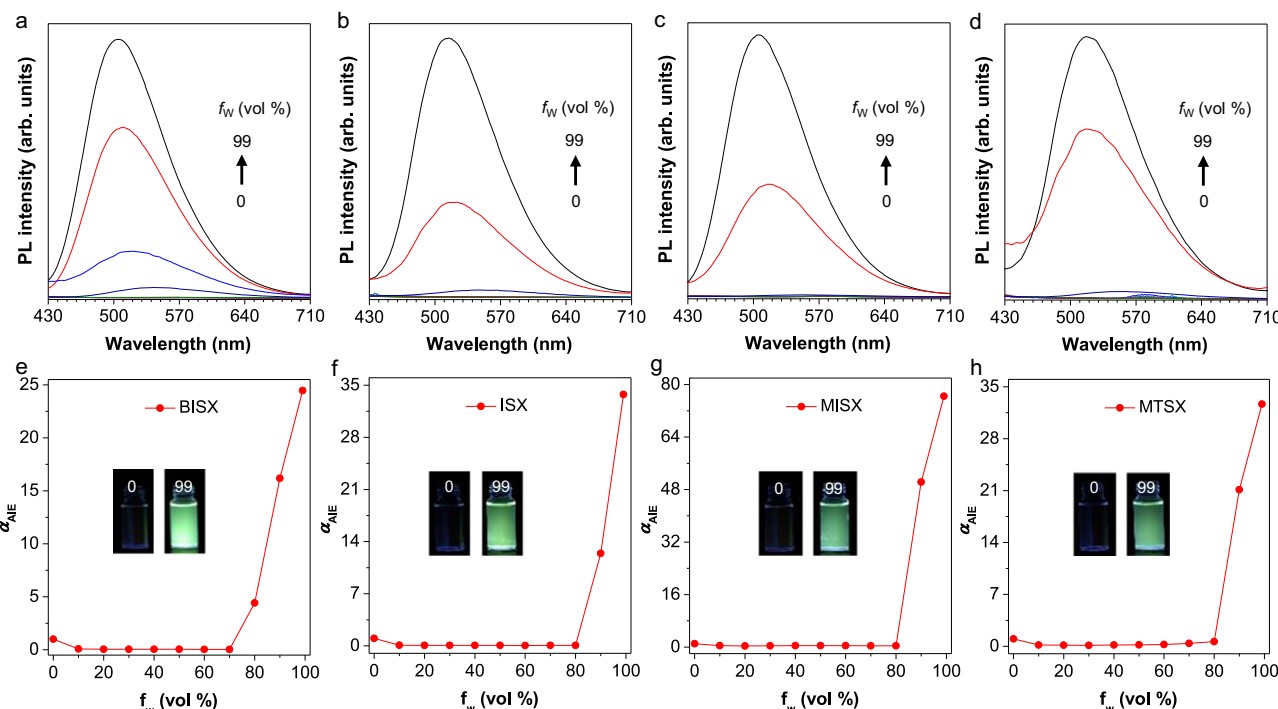

**Fig. 2 | AIE features of BISX, ISX, MISX and MTSX. a−d** PL spectra of BISX, ISX, MISX and MTSX in THF/water mixtures with different water fractions ($f_w$). Concentration ($c$) = 1 × 10$^{-5}$ M, excitation wavelength ($\lambda_{ex}$) = 365 nm. **e−h** Plots of relative PL intensity $\alpha_{AIE}$($I/I_O$) versus $f_w$, where $I_O$ was the maximal PL intensity in THF solution. Inset: fluorescence images of BISX, ISX, MISX and MTSX in THF/water mixtures with $f_w$ = 0% and 99%, respectively.

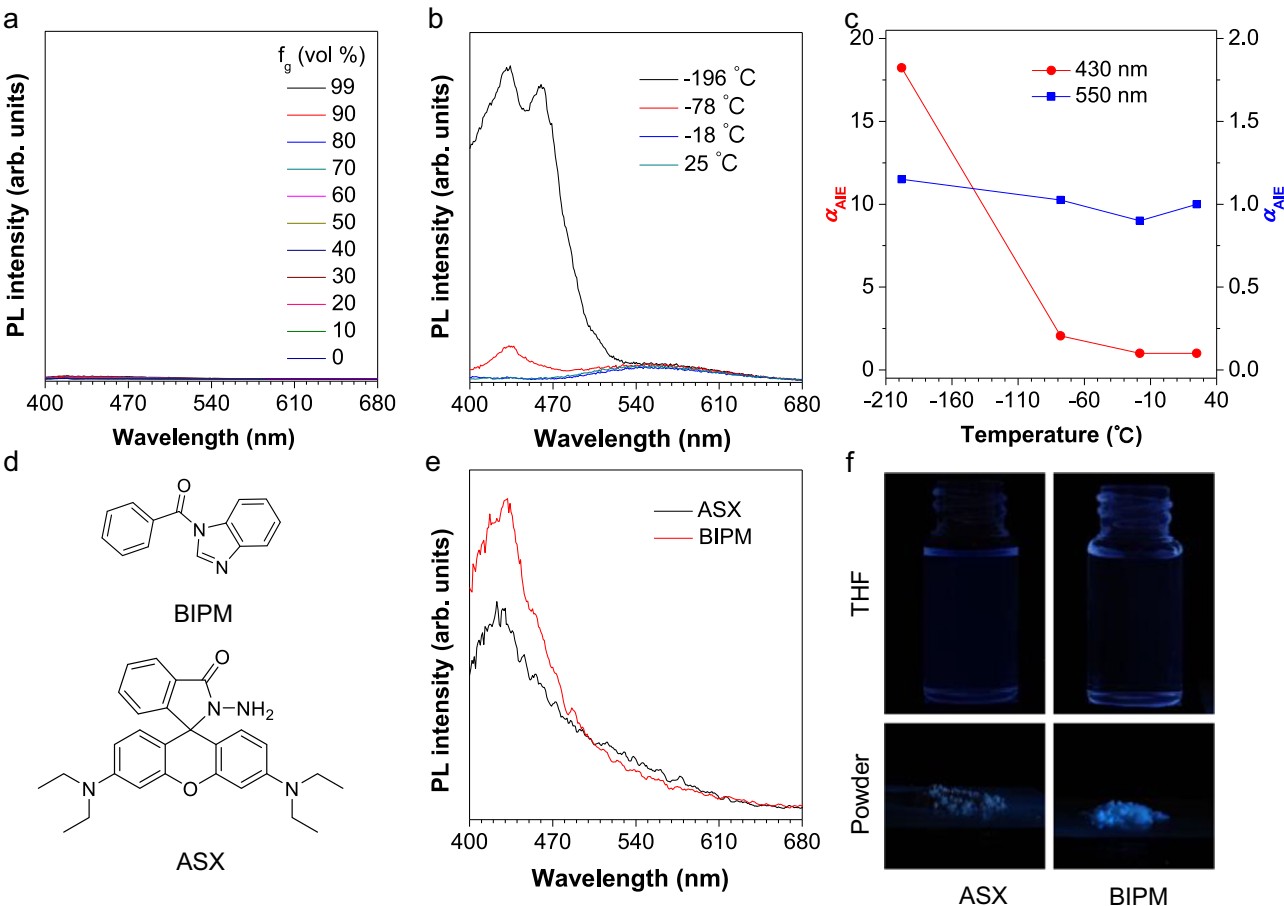

**Fig. 3 | Viscosity- and temperature-dependent experiments. a** PL spectra of BISX in methanol/glycerol mixtures with different glycerol fractions ($f_g$). Concentration ($c$) = $1 \times 10^{-5}$ M, excitation wavelength ($\lambda_{ex}$) = 365 nm. **b** PL spectra of BISX in THF solution with different temperatures. $c$ = $1 \times 10^{-5}$ M, $\lambda_{ex}$ = 365 nm. **c** Plots of relative PL intensity $\alpha_{AIE}$ ($I/I_0$) versus temperature. **d** Chemical structures of BIPM and ASX. **e** PL spectra of BISX, ASX, and BIPM in THF solution. $c$ = $1 \times 10^{-5}$ M, $\lambda_{ex}$ = 365 nm. **f** Fluorescence images of ASX and BIPM in the solution and aggregate state taken under 365 nm UV irradiation.

behavior of this non-typical AIE system. As a result, ASX, an analog of xanthene, was designed and prepared from rhodamine B (Fig. 3d, Supplementary Fig. 24a, Supplementary Figs. 25–27). A series of compounds, namely BIPM, IPM, MIPM and MTPM, corresponding to the isoquinolinone part of BISX, ISX, MISX and MTSX were also synthesized, respectively (Fig. 3d, Supplementary Fig. 24b, Supplementary Figs. 28–39). When checking the PL performance of ASX and BIPM, we found both compounds were weakly emissive in the solution and aggregate state, suggesting that the AIE property of BISX originates from neither the upper xanthene core nor the bottom nitrogen heterocyclic part (Fig. 3f, Supplementary Fig. 40). It was found that both the emission peaks of ASX and BIPM in THF solutions locate at 430 nm, corresponding to the enhanced emission peak of BISX in THF at low temperature, indicating that decreasing temperature restricted the motion of single molecular species and benefited the emission of the localized emission of upper (isoquinolinone) or bottom (xanthene) moieties (Fig. 3e). Same conclusions were also drawn for compounds ISX, MISX and MTSX (Supplementary Fig. 40). Then we assumed that their AIE behavior may result from the chromophore generated by intermolecular through-space interactions after aggregation. Excitation spectra of the four compounds were measured, respectively (Supplementary Fig. 41–44). For BISX, one set of peaks appeared between 270 nm and 340 nm, corresponding to its absorption spectrum in Supplementary Fig. 41. Besides, different from its absorption, a weak and broad excitation peak between 350 nm and 400 nm was observed, suggesting some unfavored transition processes may exist

within these molecules. The excitation behaviors of ISX, MISX and MTSX were similar to BISX.

To unveil whether the intermolecular through-space interaction is responsible for the aggregate emission of these closed-form rhodamine derivatives, we analyzed the single-crystal structures of the four compounds. The single crystals of BISX, ISX and MTSX were obtained by slow evaporation of their acetonitrile solutions, respectively, while the single crystal of MISX was grown from its acetonitrile-water (4:6, v/v) mixtures. For BISX crystal, except for short contacts including hydrogen bonding and C-H···π interactions, π-π interactions at the distance of 3.359 Å were also found between the benzo-imidazole rings (Fig. 4a and Supplementary Fig. 45). These intermolecular interactions may facilitate the restriction of molecular motion of BISX, thus inhibiting the nonradiative decay. And intermolecular π-π interactions may benefit the formation of the chromophore to induce its AIE property. Interestingly, ISX crystal, with only hydrogen bonds (2.340 Å and 2.976 Å) and C-H···π interactions (2.837 Å and 2.806 Å) rather than π-π interactions, also exhibited similar AIE behavior like BISX (Supplementary Fig. 46 and Fig. 4b). As for MISX crystal, C-H···π interactions with the distance of 2.892 Å were observed (Supplementary Fig. 47). Though hydrogen bonding interactions (C-H···O) with the distance of 2.596 Å between two isoquinolinone rings could be found, there was no observable π−π interaction. Staggered packing was observed for the two rings, similar to ISX (Fig. 4c). MTSX crystal also possessed π-π interactions, analogous to BISX (Fig. 4d and Supplementary Fig. 48). Therefore, comparing the packing mode of BISX and

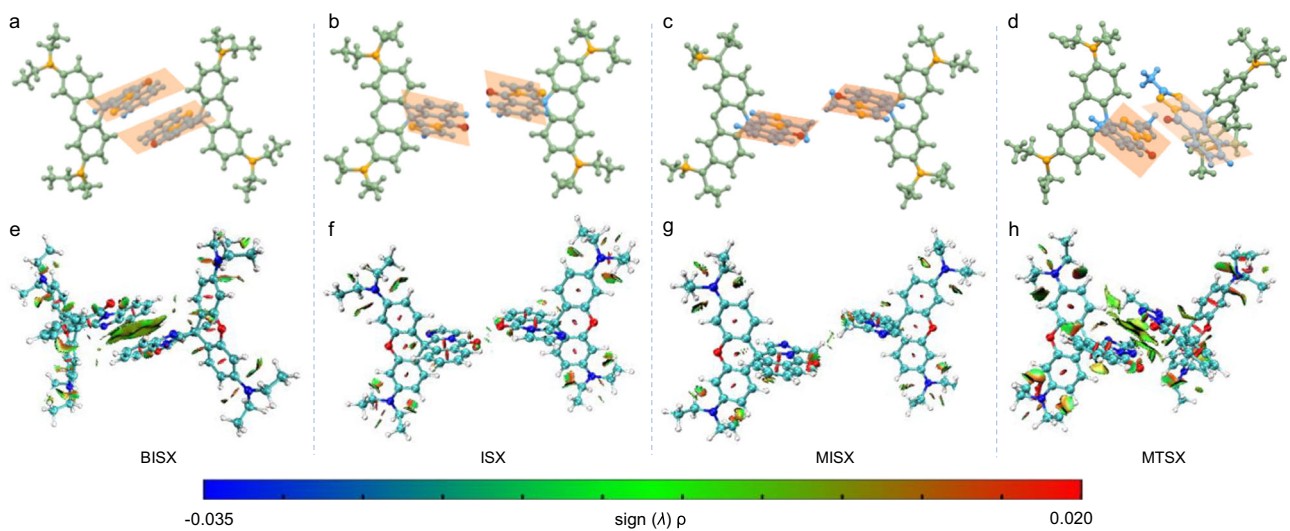

**Fig. 4 | Calculations on intermolecular interactions.** Single-crystal structures (**a**–**d**) and reduced density gradient (RDG) analysis (**e**–**h**) for dimers of BISX, ISX, MISX, and MTSX, respectively. (RDG = 0.5).

ISX crystal, and MISX and MTSX, we concluded that intermolecular through-space interaction did not necessitate the chromophore formation and the AIE properties of these rhodamine derivatives. To further verify our proposal, non-covalent interactions were calculated by reduced density gradient (RDG) method[40]. As shown in Fig. 4e–h, substantial intermolecular through-space interactions were found for BISX and MTSX, while ISX and MISX exhibited negligible intermolecular interactions. Thus, combining the crystal structure analysis and RDG calculation results, we could conclude that intermolecular through-space interaction is not the fundamental factor in inducing AIE behavior. Since the intermolecular through-space interaction was excluded, we would like to explore other kinds of possibilities that are responsible for the chromophore formation and AIE behavior of these closed-form rhodamines. We doped BISX, ISX, MISX and MTSX into PMMA films with different ratios to restrict their molecular motion, and measured their luminescence properties subsequently (Supplementary Fig. 49 and Fig. 5). For BISX, emission intensity of the doping system decreased with a reduced doping ratio from 5% to 0.1% (Fig. 5a). Similar results were also found for ISX, MISX and MTSX doped in PMMA film (Fig. 5b–d). This suggested that the emission of the four compounds in the isolated state was kind of forbidden even if they were confined in a rigid matrix, while the aggregation at high concentration can effectively break the transition forbidden to light up the emission. For conventional AIEgens, it is known that both rigidification in the polymeric matrix and reducing environmental temperature can promote the radiative decay rate since molecular motion is restricted, thus providing strong emission. Thus, combined with the previous cooling experimental results, we assumed that except for the RIM mechanism, there may be other reasons for the AIE properties.

Although the four compounds were nonconjugated with donor (D) xanthene and acceptor (A) isoquinolinone parts isolated by an sp[3] carbon atom, intramolecular charge transfer may exist in a through-space way to narrow the band gap, facilitating the chromophore formation. This hypothesis was further supported by theoretical calculation since the highest occupied molecular orbitals (HOMOs) were distributed on the D moieties (xanthene) while the lowest unoccupied molecular orbitals (LUMOs) were generally distributed on the A moieties (isoquinolinone) as shown in Supplementary Fig. 50. Take BISX for instance, in dilute solutions, the neighboring D and A groups were almost perpendicular to each other with a dihedral angle of 89.5° in the ground state (Fig. 6a). Thus, electron clouds of the isolated conjugation rings were not overlapped at all, which has been well recognized

as a forbidden factor for electron transition[41]. The calculated small oscillator strength value of 0.0042 also verified that (Supplementary Table 3). In the excited state, a relatively smaller dihedral angle of 77.73° was found for the optimized geometry which might allow the electron transition. While in the ground state, such large torsion from 89.5° to 77.73° was hard to realize, and the absorbed energy would undergo a non-radiative decay pathway by structural relaxation. Further calculations proved that change of electron clouds mainly took place on A moiety of BISX (CT$_A$), almost the same with that of isolated isoquinolinone molecule from S$_1$ → S$_0$ (Supplementary Fig. 51). Thus, in the solution state, the system was almost non-emissive, most excitons will come to the ground state from the CT$_A$ state rather than charge transfer from D to A (CT$_{D-A}$) excited state, making CT$_{D-A}$ become a dark state (Fig. 6a). This diagram also well accounted for the emission behavior of BISX in the previous cooling experiments (Fig. 3b–c). At room temperature, due to the flexible molecular motion, the absorbed energy was mainly dissipated via the nonradiative decay pathway, affording negligible CT$_A$ emission. And the weak CT$_{D-A}$ emission which is responsible for the AIE peak was ascribed to a dark state. Although reducing temperature to restrict the molecular motion benefits enhancing the CT$_A$ emission at 430 nm, it affects scarcely the forbidden CT$_{D-A}$ emission of the perpendicular D-A structure. However, the aggregation seems to break the transition forbidden of an isolated state since these lightless nonconjugated rhodamine single species emit brightly upon aggregation or crystallization. What happened during the aggregation? Which kind of factors breaks the transition forbidden of molecular species?

To gain deep insight into these questions, theoretical calculations based on single-crystal structures were carried out. In BISX crystals, different from its orthogonal structure in the solution state, a smaller D-A dihedral angle of 83.66° within the crystal was observed. Unlike the freedom state in which BISX exhibit perpendicular orthogonal structure with almost non-overlapped HOMO and LUMO (Supplementary Fig. 52), obvious electronic structure change of BISX molecule in aggregate state was observed with more effective HOMO and LUMO overlap. The improved HOMO and LUMO overlap benefits effective CT$_{D-A}$ transition to undergo radiative decay. Therefore, the aggregation not only helps break the transition forbidden in the solution state but facilitates the allowed CT$_{D-A}$ transition in aggregate state (Fig. 6b and Supplementary Fig. 54). Based on the single-crystal structure, QM/MM calculations were further performed by the ONIOM model in the Gaussian 16 package

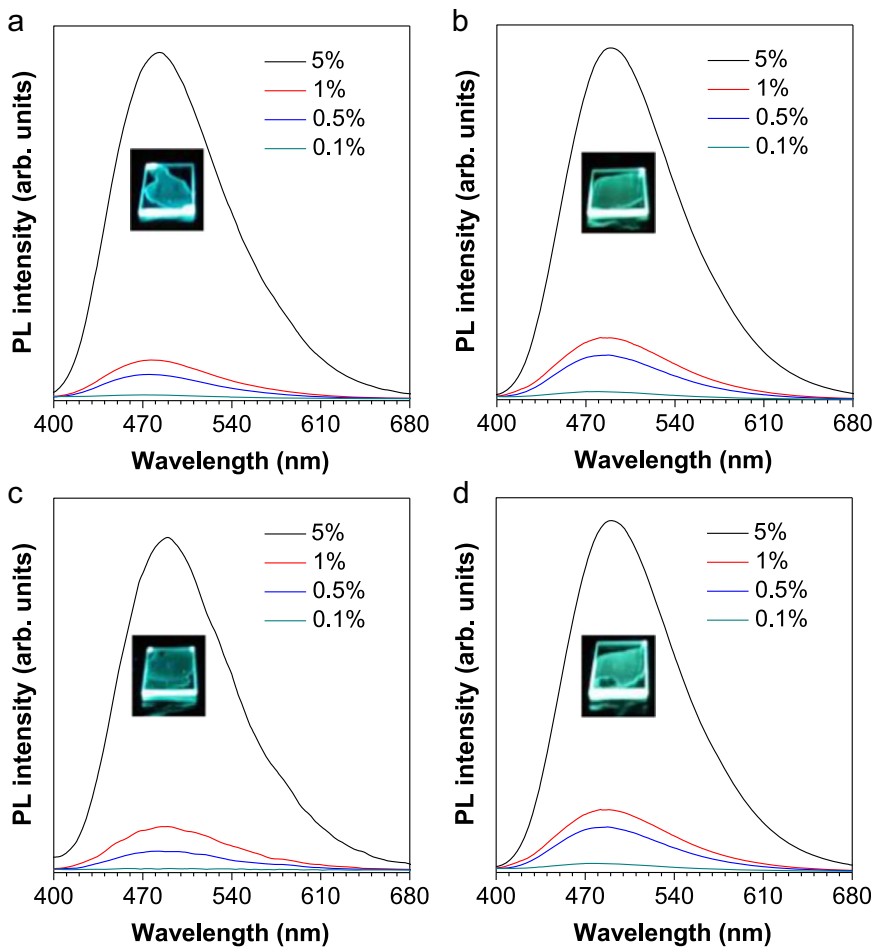

**Fig. 5 | Doping BISX, ISX, MISX and MTSX into PMMA films with different ratios.** PL spectra of BISX, ISX, MISX and MTSX (**a**–**d**) in PMMA film with different weight ratios (m/m), respectively. Inset: fluorescence images of the corresponding compounds in PMMA (5%). Excitation wavelength ($\lambda_{ex}$) = 365 nm.

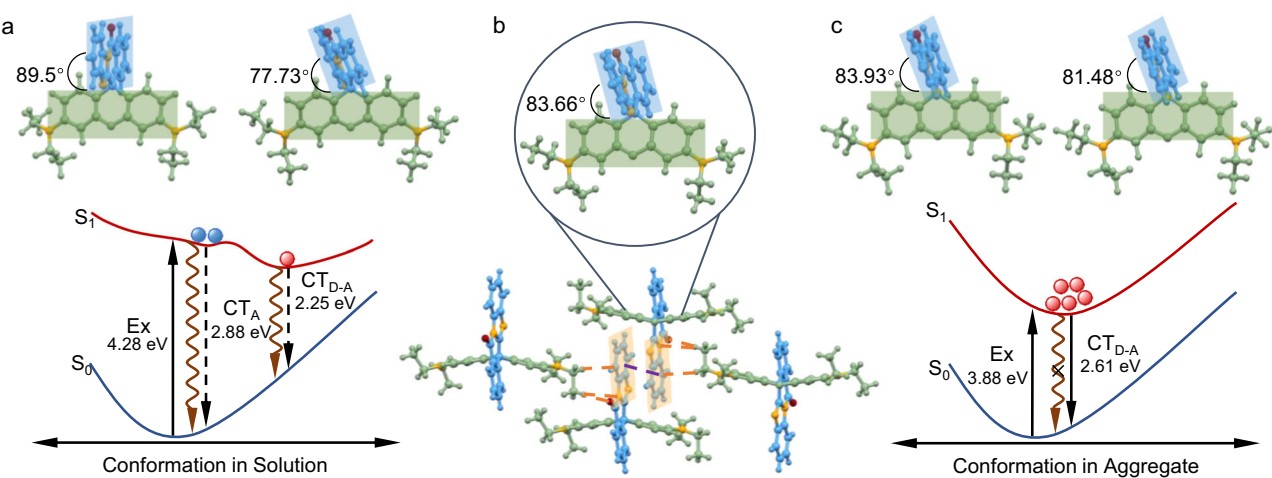

**Fig. 6 | Mechanistic illustration of the emission behavior of BISX in different states. a** Potential energy surface of BISX in the THF solution and the optimized ground-state and excited-state geometries. **b** Single-crystal structure and packing arrangement of BISX. **c** Potential energy surface of BISX in the crystalline state and the optimized ground-state and excited-state geometries.

(Supplementary Fig. 55). As shown in Fig. 6c, the variation of the dihedral angle between the excited state and ground state was merely 2.45°, leading to more accessible structural relaxation to the geometry of allowed $CT_{D-A}$ transition. Thus, in the aggregate state,

BISX displayed $CT_{D-A}$ emission at 475 nm. And the obtained energy gap between $S_1$ and $S_0$ of $CT_{D-A}$ (2.61 eV) is larger than that of in the solution state (2.25 eV), making $CT_{D-A}$ in the aggregate state closer to the Franck-Condon geometry. And the trend observed in the

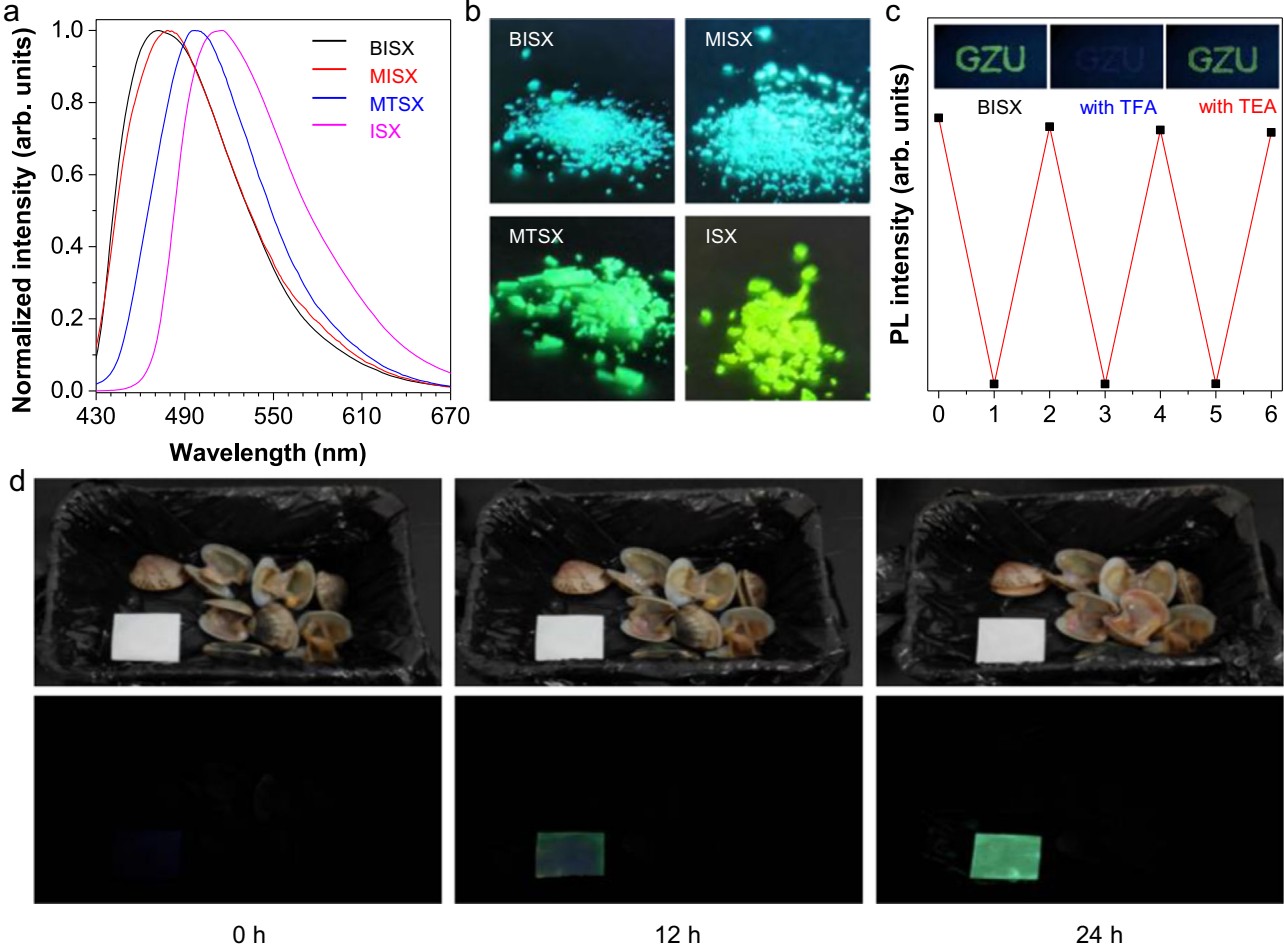

**Fig. 7 | Dynamic responses of BISX to external stimuli.** Normalized PL spectra (**a**) and fluorescence images (**b**) of BISX, ISX, MISX and MTSX in the aggregate state. Excitation wavelength ($\lambda_{ex}$) = 365 nm. **c** Maximal PL intensity of BISX treated with trifluoroacetic acid (TFA) and triethylamine (TEA). Inset: fluorescence images taken under 365 nm UV irradiation, respectively. **d** Spoilage detection of clams in sealed packages for 24 h at room temperature using BISX. Photographs taken under daylight (upper) and 365 nm UV irradiation (bottom).

calculated data (Supplementary Table 4) also agrees with the experimental results. Compared to the free molecules with high rotational and translational flexibility in the solution state, molecules within the confined aggregate state were interfered by various interactions (C-H···π, π-π, hydrogen bonding), leading to conformation change. Thus, the conformation change in the aggregate state further boosted the transition allowed $CT_{D-A}$ emission, affording the AIE feature of these nonconjugated molecules.

**Application in the detection of released amines**

Additionally, we checked the PL performance of these nonconjugated rhodamine derivatives in the solid state, respectively. As shown in Fig. 7a, b, BISX, ISX, MISX and MTSX crystals displayed bright green-blue color fluorescence with emission peaks at 475 nm, 515 nm, 478 nm, and 497 nm, respectively. Their quantum yields were measured as 64.1%, 46.0%, 69.4%, and 72.7%, which are ultrabright visible emissions for these unorthodox nonconjugated structures (Supplementary Table 2). Notably, we found that the bright fluorescence of BISX was quenched completely when exposed to trifluoroacetic acid (TFA). And the emission recovered when the acid-treated sample was exposed to triethylamine (TEA), as shown in Supplementary Fig. 54a. We assumed introducing TFA protonated the nitrogen atom of BISX (HBISX⁺), while later TEA treatment made it deprotonation. When further checking the chemical structure of

BISX, there are two nitrogen atoms contained in the conjugation backbone (Supplementary Fig. 54b). The lone pair electrons of the sp³ nitrogen are perpendicular to the plane of the aromatic ring, forming lone-pair π conjugation, thus the sp³ N is difficult to be protonated. For the sp² N, in the contrast, its lone pair does not participate in conjugated systems, as they are parallel to the molecular plane. The sp² hybridized nitrogen is more likely to be protonated. The calculated electron density maps based on electrostatic potential further verified our hypothesis, by giving the result that sp² N is more electron-rich than sp³ N and the diethylamine substituents (Supplementary Fig. 54c). To further prove it, we calculated the frontier molecular orbitals of protonated compounds of BISX based on their optimized ground-state geometries (Supplementary Fig. 55). It was found that the protonated structure with a protonated diethylamine substituent is unstable, exhibiting a higher electronic energy of 17.7 kcal/mol compared to the protonated structure with a protonated sp² N. We then tested the reversibility and reliability of the system by alternatively exposing the BISX film to TFA and TEA vapors (Fig. 7c). The bright and dark states could be converted for more than three consecutive cycles with negligible fatigue, revealing the high reversibility and repeatability of this sensor system. Inspired by its "turn-on/ off" response to ammonia vapor after being post-treated with acid, we employed BISX as a food spoilage sensor. We bought fresh clams, crayfish, and fish from the seafood market as

detection samples. The fresh seafood was sealed in black-packaged containers with TFA-treated thin films of BISX, and kept at room temperature for 24 h, respectively. After 12 h, faint green emission was observed. After 24 h treatment, the emission became much stronger (Fig. 7d). For crayfish and fish, as shown in Supplementary Figs. 56 and 57, the thin film gave a strong green fluorescence when sealed at 12 h, indicating their faster decomposition rate compared to clams. In this regard, food spoilage sensors could be generated for sensitive detection of ammonia and biogenic vapors generated through the food spoilage process.

## Discussion

In summary, we synthesized a series of rhodamine-based compounds with cross-shaped six-membered structures, namely, BISX, ISX, MISX, and MTSX, via a feasible and simple approach based on C−N and C−C coupling cyclization reaction. The compounds displayed typical AIE behavior. While different from conventional AIEgens, their AIE emission intensity did not enhance when increasing viscosity or decreasing temperature, suggesting their AIE properties were not simply attributed to the typical RIM mechanism. Combined with analysis from single-crystal structure, optical properties investigation, and theoretical calculations, we proposed that molecular conformation change within confined microenvironment in aggregates boosted the intramolecular CT transition, which caused their luminescence variations between the solution and aggregation state. When dissolved in THF solutions, molecules in the ground state possessed a transition forbidden orthogonal structure, making it difficult to approach the transition allowed CT excited state. In the aggregate state, however, the ground-state orthogonal conformation was changed due to various intermolecular interactions, affording an initial angle for realizing the transition allowed CT excited-state conformation. Besides, taking advantage of the sensitive "turn-on/ off" emission behavior towards acid-base treatment, spoilage detection on seafood was realized. Therefore, the current work provides an insight for understanding unconventional AIE system and offers a deeper and more general understanding on the variations between single molecules and aggregates. It is poised to steer our endeavors in crafting aggregate systems that boast innovative structures and functionalities.

## Methods

### General information

All chemicals were purchased from commercial suppliers and used without further purification. The solvents were used as analytic grade during the synthesis of the probes, and the chromatographic grade solvents were used for UV−Vis, and fluorescence instrument tests, etc. Deionized distilled water was used throughout all experiments. All the reactions were magnetically stirred and heated by an oil bath (IKA RCT basic), and thin-layer chromatography (TLC) was Spectrochem GF254 silica gel coated plates. NMR spectra were measured on a JEOL−ECX 500 and Bruker Biospin AG−400 spectrometer at room temperature using TMS as an internal standard. High-resolution Mass spectrometry (HRMS) spectra were recorded on a Thermo Scientific Q Exactive Quadrupole-Orbitrap mass spectrometer. The crystal data were collected by Bruker Smart Apex II CCD diffractometer. Fluorescence spectra were performed on a PTI QuantaMaster 8000 (Horiba Trading CO., LTD). Fluorescence lifetimes were carried out with FluoroMax-TCSPC (Horiba Trading CO., LTD). Quantum efficiencies were carried out on a PTI QuantaMaster 8000 (Horiba Trading CO., LTD) using an integrating sphere apparatus. UV−Vis spectra were performed on a TU−1900 spectrophotometer. DLS was recorded on DelsaNano C (Beckman Coulter).

### Computational methods

**DFT-calculated possible mechanism of MISX reaction.** All the calculations were carried out using the Gaussian 16 program package[42]. Single-point energy calculations and all these structures were optimized using the M06-2X/6-31 G (d)[43,44] and PCM solution methods in THF for MISX.

**Solution state.** BISX, ISX, MISX and MTSX were optimized at M06-2X/6-31 G (d) and PCM solution methods in THF. Frequency calculations were performed to confirm the characteristics of all the calculated structures as minima.

**Crystal state.** BISX was optimized at M06-2X/6-31 G (d) in gas. Frequency calculations were performed to confirm the characteristics of all the calculated structures as minima. Subsequently, (TD) M06-2X /6-31 G(d) in gas under the Gaussian 16 package.

**Aggregate state.** We used combined quantum mechanics and molecular mechanics methods (QM/MM). An ONIOM model was constructed by cutting a cluster containing 54 BISX molecules from the single crystal structure to calculate by (TD)M06-2X/6-31 G(d) methods in gas under the Gaussian 16 package.

## Data availability

The authors declare that all the data supporting the findings of this manuscript are available within the manuscript and Supplementary Information files and available from the corresponding authors upon request. The X-ray crystallographic coordinates for structures reported in this study have been deposited at the Cambridge Crystallographic Data Centre (CCDC) under deposition numbers 2051533 (BISX), 2051534 (ISX), 1991171 (MISX), and 2051536 (MTSX). These data can be obtained free of charge from The Cambridge Crystallographic Data Centre via www.ccdc.cam.ac.uk/data_request/cif.

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

## Acknowledgements

The authors are grateful for the funding support from the National Natural Science Foundation of China (21788102, 32372610, 52003228, 52273197, 52333007), the Research Grants Council of Hong Kong (C6014-20W), the Innovation and Technology Commission (ITC-CNERC14SC01), the Guizhou Provincial S&T Project (2018[4007]), Program of Introducing Talents of Discipline to Universities of China (D20023, 111 Program), Science, Technology and Innovation Commission of Shenzhen Municipality (JCYJ2021324134613038, GJHZ20210705141810031), Shenzhen Key Laboratory of Functional Aggregate Materials (ZDSYS20211021111400001, KQTD20210811090142053), the Open Fund of Guangdong Provincial Key Laboratory of Luminescence from Molecular Aggregates, South China University of Technology, Guangzhou 510640, China (2019B030301003), and the Project funded by China Postdoctoral Science Foundation (2022M72308), National Key Research and Development Program of China (2022YFD1700300).

## Author contributions

L.-L.Y. and H.W. contribute equally to this work. L.-L.Y., H.W., Z.Z., S.Y., and B.Z.T. conceived and designed the experiments. L.-L.Y. and H.W. performed the synthesis, performed the single-crystal measurements, photophysical measurements, and analyzed the data. Q.L., J.-Y.C., and A.-L.T. contributed to the preparation of the figures. L.-L.Y, B.W., and J.Z. conducted theoretical calculations. L.-L.Y., H.W., Q.L., J.W.Y.L., Z.Z., S.Y., and B.Z.T. took part in the discussion and gave important suggestions. L.-L.Y., H.W., and Q.L. wrote the paper with the help of Z.Z. All authors approved the final version of the manuscript.

## Competing interests

The authors declare no competing interests.
