## [Peer Review File · Nature Communications]

Understanding the AIE phenomenon of nonconjugated rhodamine derivatives via aggregation-induced molecular conformation changeReviewers' Comments:

Reviewer #1:

Remarks to the Author:

The manuscript by Tang et al. reported a family of nonconjugated closed-form rhodamine derivatives as AIE materials and disclosed the underlying mechanism for their AIE behavior. The authors claimed that symmetry breaking-boosted intramolecular charge transfer emission during the state transformation is the key to promote the AIE performance. This elucidation is novel and different from conventional AIE mechanism based on restriction of intramolecular motion. The conclusion is verified by detailed experimental and theoretical studies and should be inspiring in the field of luminescent materials. The authors also demonstrated an important sensing application based on the fluorescence turn-on character of the AIE dyes. Generally, the work deserves publication in Nature Communications after some minor revisions:

1. In scheme 2, the color of the chemical structure of Rhodamine B-based AIEgens was ambiguous, and a correct caption indicating its state should be provided, such as in solid form to match Figure 6.
2. The molecules showed high QY, please provide the absorption coefficient of the molecules.
3. In page 17 line 294, the authors should revise "change of molecular structure" into "change of molecular conformation", since there was no change in the molecular composition. In page 17 line 295, "structure" should also be replaced by "conformation".
4. In page 3 line 33, "agregation" should be corrected into "aggregation". The authors should check this type of typos carefully.
5. In page 4 line 70, the phrase "in the fluorescence area" should be changed into "in the field of fluorescence".
6. In page 22 line 400, reference 1 does not contain page number; In page 23 line 424, reference 12 abbreviation of journal should be spaced.
7. The formats of λ_{ex} should be unified in all figure captions.
8. Lifetime values of the AIE dyes should be provided to better understand the origin of the emission peaks.

Reviewer #2:

Remarks to the Author:

This manuscript investigates aggregation induced emission (AIE) in rhodamine B derivatives. The authors describe their findings in term of "symmetry breaking boosted intramolecular charge transfer", and claim that their work "offers a deeper and more general understanding on the variations between single molecules and aggregates". While I fully appreciate the importance of AIE, the role of symmetry breaking, as described by the authors, is not clear and I find the proposed idea not helpful to understand the effect of aggregation. I appreciate that the explanation of AIE in this particular system is not straightforward, but the explanation provided by the authors is not satisfactory and the manuscript is not ready for publication in a high profile journal. I recommend rejection. See detailed comments.

(1) The meaning of the term 'symmetry' as used by the authors is not clear because it is ambiguous, and this makes understanding the general idea of the paper difficult. In the context of Landau theory, the 'symmetry breaking' that occurs during the phase transition probably refers to the transition from an isotropic to an anisotropic environment. However, the term symmetry has other uses that are not consistent with the one in Landau theory, eg crystalline systems are more symmetric than solutions considering that they have crystallographic point groups. This makes the term ambiguous and not very useful. In addition, whatever name one might give to it, the phenomenon described by Landau theory occurs for any system undergoing aggregation, ie it might be invoked to explain AIE in any AIEgen and not just the current one. It is not clear what particular role it plays here.

Going deeper into the specific explanation offered by the authors, their arguments refer to the structural changes that the molecule undergoes in the ground and excited states, both in solution and in the crystal. In fact, this mechanistic explanation lies perfectly in line with the RIM model, the

difference with other systems being that here the structural differences between solution and crystal are quite small. These small differences might explain, for instance, the lack of temperature and viscosity effects. Therefore I do not see how the ambiguous 'symmetry breaking' idea contributes to understanding AIE in this system.

(2) The mechanistic hypotheses need more support from the calculations. In figure 5a, the authors invoke two excited states, LE and CT. What is the evidence for that? The orbitals in Figure S48 suggest that S1 is a CT state, why do the authors claim that emission in solution occurs from the LE state? The insight one can obtain from the frontier molecular orbitals provided in the manuscript is limited, a more complete study is required to obtain more insights into the mechanism.

(3) Regarding the measured excited-state life times in Figures S17-S20, the authors argue that the life times in the crystal are longer by one order of magnitude, but the curves shown for the crystal have a very short, probably sub-ns component that seems to have been neglected (eg in figure 17b, the intensity has a sharp initial drop from approx. 10^4 to $5 \cdot 10^2$). Has this initial drop been included in the fitting? Can it be explained? Please also explain what is exactly measured in these figures (eg transient absorption, fluorescence, etc).

(4) Turning to the application, the authors state in line 314 that in acid media, the sp² hybridized nitrogen is more likely to be protonated than the sp³ one. However, the diethylamine substituents of the dibenzopyran moiety are expected to be even more basic than the sp² nitrogen. How does this affect the proposed sensor mechanism?

Point-by-point Response to the Reviewers' Comments

Reviewer 1:

The manuscript by Tang et al. reported a family of nonconjugated closed-form rhodamine derivatives as AIE materials and disclosed the underlying mechanism for their AIE behavior. The authors claimed that symmetry breaking-boosted intramolecular charge transfer emission during the state transformation is the key to promote the AIE performance. This elucidation is novel and different from conventional AIE mechanism based on restriction of intramolecular motion. The conclusion is verified by detailed experimental and theoretical studies and should be inspiring in the field of luminescent materials. The authors also demonstrated an important sensing application based on the fluorescence turn-on character of the AIE dyes. Generally, the work deserves publication in Nature Communications after some minor revisions:

Response: We sincerely thank the reviewer for carefully reviewing our manuscript and positive comments and for raising these valuable questions and suggestions that help us to make improvement.

1. In scheme 2, the color of the chemical structure of Rhodamine B-based AIEgens was ambiguous, and a correct caption indicating its state should be provided, such as in solid form to match Figure 6.

Response: Thank you so much for the valuable comment. After careful checking, we found that the PL spectra and photos of MISX and ISX in Figure 6a and 6b was mislabeled. After corrections, the color of each compound in Figure 2 agrees with that in Figure 6.

On page 7 of the revised manuscript,

Scheme 1. (a) Previously reported work on Rhodamine B-based AIEgens via introducing rotors. (b) Unique rotor-free Rhodamine B-based AIEgens (BISX, ISX, MISX and MTSX) reported in this work. **The filled colors indicate their crystal emission, respectively.**

On page 19 of the revised manuscript,

Figure 6. (a) Normalized PL spectra (a) and fluorescence images (b) of BISX, ISX, MISX and MTSX in the aggregate state. λ_{exc} = 365 nm. (c) Maximal PL intensity of BISX treated with trifluoroacetic acid (TFA) and triethylamine (TEA). Inset: fluorescence images taken under 365 nm UV irradiation, respectively. (d) Spoilage detection of clams in sealed packages for 24 h at room temperature using BISX. Photographs taken under daylight (upper) and 365 nm UV irradiation (bottom).

2. The molecules showed high QY, please provide the absorption coefficient of the molecules.

Response: Thank you so much for the valuable comment. We have measured the absorption coefficient of each compound: $7446 \text{ M}^{-1}\cdot\text{cm}^{-1}$ for BISX, $7146 \text{ M}^{-1}\cdot\text{cm}^{-1}$ for ISX, $7700 \text{ M}^{-1}\cdot\text{cm}^{-1}$ for MISX, and $7566 \text{ M}^{-1}\cdot\text{cm}^{-1}$ for MTSX, respectively.

On page 8 of the revised manuscript,

“As shown in Figure S14, the four compounds displayed similar characteristic absorption spectra with several sharp bands between 240 nm to 330 nm in THF solutions, which could be attributed to their intramolecular $n\text{-}\pi^*$ transition and $\pi\text{-}\pi^*$ transition. The molar absorption coefficient of BISX, ISX, MISX and MTSX is measured as $7446 \text{ M}^{-1}\cdot\text{cm}^{-1}$, $7146 \text{ M}^{-1}\cdot\text{cm}^{-1}$, $7700 \text{ M}^{-1}\cdot\text{cm}^{-1}$, and $7566 \text{ M}^{-1}\cdot\text{cm}^{-1}$, respectively.”

On page 14 of the revised SI,

Figure S14 UV-vis absorbance spectrum of BISX, ISX, MISX, and MTSX in THF. $c = 1 \times 10^{-5} \text{ M}$.

3. In page 17 line 294, the authors should revise “change of molecular structure” into “change of molecular conformation”, since there was no change in the molecular composition. In page 17 line 295, “structure” should also be replaced by “conformation”.

Response: Thank you so much for the valuable comment. We have made corrections in the revised manuscript.

On page 17 of the revised manuscript

“.....molecules within the confined aggregate state were interfered by various interactions (C-H···π, π-π, hydrogen bonding), leading to conformation change. Thus, the conformation change in the aggregate state.....”

4. In page 3 line 33, “agregation” should be corrected into “aggregation”. The authors should check this type of typos carefully.

Response: Thank you so much for pointing out the typos. We have made corrections in the revised manuscript, and carefully went through the manuscript.

On page 3 of the revised manuscript,

“In this work, by taking the unique closed-form rhodamines-based aggregation-induced emission (AIE) system as model compounds,”

5. In page 4 line 70, the phrase “in the fluorescence area” should be changed into “in the field of fluorescence”.

Response: Thank you so much for the valuable comment. According to the suggestions of reviewer 2, we have re-composed the introduction part and used the expression you suggested “in the field of XX”.

On page 4 of the revised manuscript,

“Furthermore, in the field of biology, the folded 3D structure of.....”

6. In page 22 line 400, reference 1 does not contain page number; In page 23 line 424, reference 12 abbreviation of journal should be spaced.

Response: Thank you so much for the valuable comment. We have corrected ref 1 in the revised manuscript. And according to the suggestions of reviewer 2, we have re-composed the introduction part and deleted some references (including ref 12). We have carefully gone through the reference part.

On page 22 of the revised manuscript,

“1. Ye, S. et al. Machine learning-assisted exploration of a versatile polymer platform with charge transfer-dependent full-color emission. *Chem* **9**, 924-947 (2023).”

7. The formats of λ_{ex} should be unified in all figure captions.

Response: Thank you so much for the valuable comment. We have unified them in the revised manuscript.

On page 14 of the revised manuscript,

“Figure 4. PL spectra of BISX, ISX, MISX and MTSX (a-d) in PMMA film with different weight ratios (m/m), respectively. Inset: fluorescence images of the corresponding compounds in PMMA (5%). $\lambda_{ex} = 365 \text{ nm}$.”

On page 19 of the revised manuscript,

“Figure 6. (a) Normalized PL spectra ... $\lambda_{ex} = 365 \text{ nm}$”

8. Lifetime values of the AIE dyes should be provided to better understand the origin of the emission peaks.

Response: Thank you so much for the valuable comment. We have remeasured the lifetime of each compound.

On page 15-16 of the revised SI,

Figure S17 (a) Time-resolved fluorescence decay curves of BISX in THF at 550 nm, $c = 1 \times 10^{-5} \text{ M}$. (b) Time-resolved fluorescence decay curves of BISX crystal at 475 nm.

Figure S18 (a) Time-resolved fluorescence decay curves of ISX in THF at 550 nm, $c = 1 \times 10^{-5} \text{ M}$. (b) Time-resolved fluorescence decay curves of ISX crystal at 515 nm.

Figure S19 (a) Time-resolved fluorescence decay curves of MISX in THF at 550 nm, $c = 1 \times 10^{-5}$ M. (b) Time-resolved fluorescence decay curves of MISX crystal at 478 nm.

Figure S20 (a) Time-resolved fluorescence decay curves of MTSX in THF at 550 nm, $c = 1 \times 10^{-5}$ M. (b) Time-resolved fluorescence decay curves of MTSX crystal at 497 nm.

Reviewer 2:

This manuscript investigates aggregation induced emission (AIE) in rhodamine B derivatives. The authors describe their findings in term of "symmetry breaking boosted intramolecular charge transfer", and claim that their work "offers a deeper and more general understanding on the variations between single molecules and aggregates". While I fully appreciate the importance of AIE, the role of symmetry breaking, as described by the authors, is not clear and I find the proposed idea not helpful to understand the effect of aggregation. I appreciate that the explanation of AIE in this particular system is not straightforward, but the explanation provided by the authors is not satisfactory and the manuscript is not ready for publication in a high profile journal. I recommend rejection. See detailed comments.

Response: We thank you very much for your precious time and comments to review our manuscript and are grateful for your appreciation on the importance of AIE. We admit that our previous explanation of AIE in this particular system is not straightforward for readers to understand. According to your professional and constructive comments, we had a careful discussion and made extra experiments. We reposed the working mechanism of these AIEgens as molecular conformation change induced by aggregation, which boosted the intramolecular CT transition in the aggregate state. We also re-structured the introduction part, and updated the title and TOC graph.

1. The meaning of the term 'symmetry' as used by the authors is not clear because it is ambiguous, and this makes understanding the general idea of the paper difficult. In the context of Landau theory, the

'symmetry breaking' that occurs during the phase transition probably refers to the transition from an isotropic to an anisotropic environment. However, the term symmetry has other uses that are not consistent with the one in Landau theory, eg crystalline systems are more symmetric than solutions considering that they have crystallographic point groups. This makes the term ambiguous and not very useful. In addition, whatever name one might give to it, the phenomenon described by Landau theory occurs for any system undergoing aggregation, ie it might be invoked to explain AIE in any AIEgen and not just the current one. It is not clear what particular role it plays here.

Going deeper into the specific explanation offered by the authors, their arguments refer to the structural changes that the molecule undergoes in the ground and excited states, both in solution and in the crystal. In fact, this mechanistic explanation lies perfectly in line with the RIM model, the difference with other systems being that here the structural differences between solution and crystal are quite small. These small differences might explain, for instance, the lack of temperature and viscosity effects. Therefore I do not see how the ambiguous 'symmetry breaking' idea contributes to understanding AIE in this system.

Response: Thank you so much for your valuable comment. After careful consideration, we agree with you that the terms used “symmetry” and “symmetry breaking” are not clear. The “symmetry breaking” in our system refers to the disorder and order extent of the system in solution and crystal. We think it could help us to get a general understanding of the property difference between different states from the level of philosophy. However, indeed, it could not provide detailed information for the mechanism of the given specific AIE system we presented. Based on these considerations, we reorganized the introduction part with more focus on the science part but less attention on the philosophy part. According to the experimental results, we concluded that the aggregation could cause a conformation change, which could further induce an intramolecular charge transfer emission. Thus, we have revised the introduction part and the mechanism discussion. We also updated the title and TOC Graph to keep consistent.

Title: Understanding the AIE phenomenon of nonconjugated rhodamine derivatives via aggregation-induced molecular conformation change

TOC graph:

On page 3-6 of the revised manuscript,

“Abstract

The bottom-up molecular science research paradigm has greatly propelled the advancement of materials science. However, some organic molecules can exhibit markedly different properties upon aggregation. Understanding the emergence of these properties and structure-property relationship has become a new research hotspot. In this work, by taking the unique closed-form

rhodamines-based aggregation-induced emission (AIE) system as model compounds, we investigated their luminescent properties and the underlying mechanism deeply from a top-down viewpoint. Interestingly, the closed-form rhodamine-based AIE system did not display the expected emission behavior under high-viscosity or low-temperature conditions. Alternatively, we finally found that the molecular conformation change upon aggregation induced intramolecular charge transfer emission and played a significant role for the AIE phenomenon of these closed-form rhodamine derivatives. The application of these closed-form rhodamine-based AIE probe in food spoilage detection was also explored.

Introduction

Unveiling the structure-property relationship of organic functional materials and proposing effective materials design principle are among the most fascinating tasks for material scientists.^{1,2,3} Current molecular design of functional materials mainly follows a bottom-up molecular science research paradigm, which emphasizes the role of molecular structure in determining the properties of materials.^{4,5,6} Under the guidance of this molecular structure dominated research paradigm, various molecular materials with new structures have been designed through an experience-based subconsciousness. Taking organic luminescent materials as an example, the introduction of conjugated rigid building blocks and the expansion of the π -conjugation are the commonly used strategies to develop new luminescent materials since they are generally thought necessary for the formation of chromophore and tuning the band gap.^{7,8,9} Another example is the preparation of chiral materials or drugs, which usually needs the incorporation of chiral factors or asymmetry catalysis to induce the generation of a chiral center.¹⁰

It is no doubt that the molecular structure dominated research paradigm has made significant contributions to the advancement of modern chemistry and materials.¹¹ However, this molecular structure oriented materials development approach also suffers some emerging limitations with the expansion of the materials scope. There are more and more cases indicated that the molecular structure alone cannot fully determine the materials properties while their aggregation modes greatly influence their macroscopic performance.^{12,13} For instance, Yuan and coworkers have discovered that the chromophore-free saccharide molecules without even one phenyl ring could exhibit strong luminescence upon aggregation, and polypeptide and some other non-conjugated polymers exhibited similar phenomenon.^{14,15} Tang and coworkers discovered another type interesting luminescence phenomenon that some propeller-shaped molecules exhibit no emission in their isolated state but show bright emission in aggregation state, which they has coined as aggregation-induced emission (AIE).^{16,17} Li and Zhang et al. reported that the achiral molecules like hexaphenylsilole and tetraphenylethene (TPE) derivatives with non-chiral centers could generate circular dichroism signals and even circularly polarized light emission in aggregates while their single molecular state didn't show any chiral signals.^{18,19,20} Furthermore, in the field of biology, the folded 3D structure of polypeptide and its wrongly folded structure-resulted aggregates could also exhibit vastly different biological activities.^{21,22} These inconsistencies between molecular properties and aggregate properties can possibly be ascribed to the complex changes occurring at both molecular level and aggregate level during the aggregation process, which unfortunately has long been ignored as the role of molecular structure is overpriced. Therefore, it is highly desirable to revisit the structure-property relationship from a top-down perspective with more attention focus on the influence brought by aggregation to the molecules.

By shifting the research interest from single molecule to aggregates at condensed state, scientists have unveiled that lots of factors induced by the multibody interactions play an important role in determining the materials performance.^{23,24,25} For example, An et al. reported that H-aggregation in molecular packing could stabilize triplet excitons to realize ultralong organic phosphorescence.²⁶ Li et al. established the one-to-one correspondence between the determinate interactions and excited triplet states, which guided the design of organic phosphorescence materials.²⁷ Barrett et al. reported that co-crystallization of two or more molecules can exhibit properties that do not belong to any individual molecules.^{28,29,30} These works also highlighted how did the crystallinity, aggregate morphology, assembly behavior that occurred among multibodies affect the materials performance.^{31,32} In addition to multibody interactions, whether the aggregation will cause the change of the characteristic of molecule itself, like the molecular conformation or molecular electronic structure, leading to emerging of new macroscopic materials performance is also an interesting topic that is worth to be investigated. However, investigation from this top-down viewpoint has seldomly been considered and there are very few related studies. Thus, it is crucial to investigate how the spatial confinement resulting from aggregation and changes in the surrounding molecular environment affect molecular characteristic like the molecular conformation. In other words, further research is needed to understand the differences in molecular conformation between the single molecular state and the aggregate state. This investigation will provide valuable insights into the influence of aggregation on materials properties and enable the development of strategies to optimize their performance.”

“Through a top-down viewpoint, we finally found that the molecular conformation change upon aggregation induced intramolecular charge transfer emission and played a significant role in the AIE phenomenon of these closed-form rhodamine derivatives.”

On page 17 of the revised manuscript,

“Compared to the free molecules with high rotational and translational flexibility in the solution state, molecules within the confined aggregate state were interfered by various interactions (C-H \cdots π , π - π , hydrogen bonding), leading to conformation change. Thus, the conformation change in the aggregate state further boosted the transition allowed CT_{D-A} emission, affording the AIE feature of these nonconjugated molecules.”

On page 19 of the revised manuscript,

“Combined with analysis from single-crystal structure, optical properties investigation, and theoretical calculations, we proposed that molecular conformation change within confined microenvironment in aggregates boosted the intramolecular CT transition, which caused their luminescence variations between the solution and aggregation state.”

2. The mechanistic hypotheses need more support from the calculations. In figure 5a, the authors invoke two excited states, LE and CT. What is the evidence for that? The orbitals in Figure S48 suggest that S1 is a CT state, why do the authors claim that emission in solution occurs from the LE state? The insight one can obtain from the frontier molecular orbitals provided in the manuscript is limited, a more complete study is required to obtain more insights into the mechanism.

Response: Thank you so much for your valuable comment. After careful thinking on your constructive comment, we found that we confused and misunderstood the concepts of LE and CT. The LE in our system was relative to the whole molecule. It referred to the electron transition occurred only at the upper part of the molecule, and CT referred to the electron transition occurred between the upper (Acceptor) and lower (Donor) parts of the molecule. Thus, to make it clearly and avoid confusion, we deleted the descriptions of “LE”, and revised the two excited states as “change of electron clouds mainly took place on A moiety (CT_A)”, and “charge transfer from D to A (CT_{D-A})”. CT_A caused the short-wavelength emission at low temperatures, while CT_{D-A} caused the long-wavelength AIE emission.

Taking BISX as an example, we tested its temperature-dependent PL spectra in THF (Figure 2b and c). At 25 °C, the compound showed a hardly noticeable emission around 540 nm. However, upon cooling to -196 °C, a significant enhancement of fluorescence peak at 410-420 nm was observed, which corresponded to the fluorescence peaks of BIPM (analogue of the upper part of BISX). Therefore, in the solution state, BISX molecules gave almost no fluorescence due to their ability to move freely at 25 °C. However, as the temperature decreased, the molecular motion was suppressed. And since the molecules were in a dispersed state with minimal intermolecular interactions, only the upper part of BISX gave the fluorescence (CT_A) due to RIM. As shown in Figure S49, we further conducted calculations on BISX and its upper part, isoquinolinone. We found that BISX exhibited CT state only in its upper part, from HOMO-3 to LUMO, which is very similar to the electron cloud distribution of isoquinolinone from S_1 to S_0 .

Figure S49. The ground state distribution of frontier molecular orbitals of BISX (top), and isoquinolinone (bottom) molecules in THF solutions.

Therefore, we assumed that in the solution state, the CT_{D-A} of BISX is forbidden due to its perpendicular D-A structure, which caused electron clouds changes mainly took place on A moiety (CT_A). While in the aggregate state, aggregation helps break the transition forbidden in the solution state, facilitating the allowed CT_{D-A} transition (Figure 5b and S50).

On page 15 of the revised manuscript,

“...Further calculations proved that change of electron clouds mainly took place on A moiety of BISX (CT_A), almost the same with that of isolated isoquinolinone molecule from $S_1 \rightarrow S_0$ (Figure S49). Thus, in the solution state, the system was almost non-emissive, most excitons will come to the ground state from the CT_A state rather than charge transfer from D to A (CT_{D-A}) excited state, making CT_{D-A} become a dark state (Figure 5a). This diagram also well accounted for the emission behavior of BISX in the previous cooling experiments (Figure 2b-c). At room temperature, due to

the flexible molecular motion, the absorbed energy was mainly dissipated via the nonradiative decay pathway, affording negligible CT_A emission. And the weak CT_{D-A} emission which is responsible for the AIE peak was ascribed to a dark state. Although reducing temperature to restrict the molecular motion benefits enhancing the CT_A emission at 430 nm, it affects scarcely the forbidden CT_{D-A} emission of the perpendicular D-A structure.”

On page 16 of the revised manuscript,

“...The improved HOMO and LUMO overlap benefits effective CT_{D-A} transition to undergo radiative decay. Therefore, the aggregation not only helps break the transition forbidden in the solution state but facilitates the allowed CT_{D-A} transition in aggregate state (Figure 5b and S50). Based on the single-crystal structure, QM/MM calculations were further performed by the ONIOM model in the Gaussian 16 package (Figure S51). As shown in Figure 5c, the variation of the dihedral angle between the excited state and ground state was merely 2.45° , leading to more accessible structural relaxation to the geometry of allowed CT_{D-A} transition. Thus, in the aggregate state, BISX displayed CT_{D-A} emission at 540 nm. Compared to the free molecules with high rotational and translational flexibility in the solution state, molecules within the confined aggregate state were interfered by various interactions (C-H $\cdots\pi$, π - π , hydrogen bonding), leading to conformation change. Thus, the conformation change in the aggregate state further boosted the transition allowed CT_{D-A} emission, affording the AIE feature of these nonconjugated molecules.”

3. Regarding the measured excited-state life times in Figures S17-S20, the authors argue that the life times in the crystal are longer by one order of magnitude, but the curves shown for the crystal have a very short, probably sub-ns component that seems to have been neglected (eg in figure 17b, the intensity has a sharp initial drop from approx. 10^4 to 5×10^2). Has this initial drop been included in the fitting? Can it be explained? Please also explain what is exactly measured in these figures (eg transient absorption, fluorescence, etc).

Response: Thank you so much for your valuable comment. After careful checking, we found that sub-ns component was neglected, and the crystal lifetimes were not fully decayed in our previous measurement. Thus, we re-measured the time-resolved fluorescence decay curves of the four compounds.

On page 15-16 of the revised SI,

Figure S17 (a) Time-resolved fluorescence decay curves of BISX in THF at 550 nm, $c = 1 \times 10^{-5}$ M. (b) Time-resolved fluorescence decay curves of BISX crystal at 475 nm.

Figure S18 (a) Time-resolved fluorescence decay curves of ISX in THF at 550 nm, $c = 1 \times 10^{-5}$ M. (b) Time-resolved fluorescence decay curves of ISX crystal at 515 nm.

Figure S19 (a) Time-resolved fluorescence decay curves of MISX in THF at 550 nm, $c = 1 \times 10^{-5}$ M. (b) Time-resolved fluorescence decay curves of MISX crystal at 478 nm.

Figure S20 (a) Time-resolved fluorescence decay curves of MTSX in THF at 550 nm, $c = 1 \times 10^{-5}$ M. (b) Time-resolved fluorescence decay curves of MTSX crystal at 497 nm.

4. Turning to the application, the authors state in line 314 that in acid media, the sp² hybridized nitrogen is more likely to be protonated than the sp³ one. However, the diethylamine substituents of the dibenzopyran moiety are expected to be even more basic than the sp² nitrogen. How does this affect the proposed sensor mechanism?

Response: Thank you so much for your valuable comment. After carefully checking the electrostatic potential of BISX (Figure S52), we found that the electronegativity for the sp² hybridized nitrogen is stronger than the nitrogen from the diethylamine substituents, so the sp² hybridized nitrogen should more easily to be protonated. For further prove it, we calculated the frontier molecular orbitals of protonated compounds of BISX based on their optimized ground-state geometries for protonated sp² hybridized nitrogen (Figure S53 a) and the protonated nitrogen from diethylamine substituents (Figure

S53 b). The unstable isomer of the protonated structure displayed a higher electronic energy of 17.7 kcal/mol than that of the stable one. Thus, we assumed that the sp^2 hybridized nitrogen should be first protonated.

On page 18 of the revised manuscript,

“The calculated electron density maps based on electrostatic potential further verified our hypothesis, by giving the result that sp^2 N is more electron-rich than sp^3 N and the diethylamine substituents (Figure S52c). To further prove it, we calculated the frontier molecular orbitals of protonated compounds of BISX based on their optimized ground-state geometries (Figure S53). It was found that the protonated structure with a protonated diethylamine substituent is unstable, exhibiting a higher electronic energy of 17.7 kcal/mol compared to the protonated structure with a protonated sp^2 N.”

On page 34 of the revised SI,

Figure S53 Frontier molecular orbitals of protonated compounds of BISX based on their optimized ground-state geometries for protonated sp^2 hybridized nitrogen (a) and the protonated nitrogen from diethylamine substituents (b).

Reviewers' Comments:

Reviewer #1:

Remarks to the Author:

The authors have revised the manuscript according to the reviewer comments carefully. The current version of manuscript can be published in Nature Communications without further change.

Reviewer #2:

Remarks to the Author:

The manuscript has been largely improved with respect to the initial version, and the idea of aggregation-induced molecular conformation change is more convincing that the initial symmetry breaking mechanism. While I think the manuscript is potentially suitable for publication, I think that the mechanistic picture summarized in Figure 5 may be improved, and more support for the mechanism from the computations should be provided. I recommend major revisions and further review.

Detailed comments:

(1) If I understand the idea correctly, the initially excited state is S1, which in the ground state conformation (with a D-A angle $>80^\circ$) is the CT_D-A state. In solution, this state switches to the CT_A state (D-A angle $< 80^\circ$), and in the aggregate phase this state switch does not occur because the folding of the upper moiety is hindered. In my opinion, Figure 5a does not reflect this situation, because the CT_D-A state should appear first along the relaxation coordinate (it is the initially excited state), and the CT_A state should appear later, according to the state switch idea (ie CT_A and CT_D-A should exchanged their position in the Figure). Similarly, in 5c the CT_D-A emission should occur nearer to the Franck-Condon geometry (S0 minimum).

(2) Still on Figure 5, the authors only provide a qualitative picture of the computations. They need to provide the quantitative data, ie the S1 and S0 energies of the different computed minima.

(3) The authors argue that the CT_A and CT_D-A states differ in the probability of electron transition. This can be easily checked on the basis of the computed oscillator strengths. These data also need to be provided and discussed.

(4) In the computational methods section, it is not clear how the structures in Figure S6 were obtained. The statement in page 21 states that single-point energy calculations were performed with M06-2X/6-31G(d), but not how the structures were optimized. Please add this information.

Minor point:

(5) In page 15, in the statement: "While in the real case, such large torsion from 265 89.5° to 77.73° was hard to realize (...)", "real case" probably means "ground state", please correct.

Point-by-point Response to the Reviewers' Comments

Reviewer 1:

The authors have revised the manuscript according to the reviewer comments carefully. The current version of manuscript can be published in Nature Communications without further change.

Response: We sincerely thank the reviewer for carefully reviewing our manuscript and positive comments and for raising these valuable questions and suggestions that help us to make improvement.

Reviewer 2:

The manuscript has been largely improved with respect to the initial version, and the idea of aggregation-induced molecular conformation change is more convincing that the initial symmetry breaking mechanism. While I think the manuscript is potentially suitable for publication, I think that the mechanistic picture summarized in Figure 5 may be improved, and more support for the mechanism from the computations should be provided. I recommend major revisions and further review.

Response: We thank you very much for your precious time and comments to review and improve our manuscript. According to your valuable suggestions below, we have improved Figure 5 and provided more computational data. The details are listed below.

Detailed comments:

1. If I understand the idea correctly, the initially excited state is S_1 , which in the ground state conformation (with a D-A angle $>80^\circ$) is the CT_{D-A} state. In solution, this state switches to the CT_A state (D-A angle $< 80^\circ$), and in the aggregate phase this state switch does not occur because the folding of the upper moiety is hindered. In my opinion, Figure 5a does not reflect this situation, because the CT_{D-A} state should appear first along the relaxation coordinate (it is the initially excited state), and the CT_A state should appear later, according to the state switch idea (ie CT_A and CT_{D-A} should exchanged their position in the Figure). Similarly, in 5c the CT_{D-A} emission should occur nearer to the Franck-Condon geometry (S_0 minimum).

Response: Thank you so much for your valuable comment. In the solution state, the initially excited state is S_1 . In the ground state conformation, where the D-A angle $> 80^\circ$, it corresponds to the CT_A state rather than the CT_{D-A} state. This is because the Acceptor part and Donor part are nearly perpendicular to each other with a dihedral angle of 89.5° , resulting in a forbidden electron transition. However, in the aggregate state, the structural differences between the ground state and excited state are small, thus leading to the occurrence of CT_{D-A} state. Therefore, we assume that in the solution state, CT_A emission will occur closer to the Franck-Condon geometry (Figure 5a), while CT_{D-A} will emission occur closer to the Franck-Condon geometry upon aggregation (Figure 5c). We have revised Figure 5 as well as the TOC graphic accordingly.

On page 2 of the revised manuscript,

TOC Graphic

On page 16 of the revised manuscript,

Figure 5. (a) Potential energy surface of BISX in the THF solution and the optimized ground-state and excited-state geometries. (b) Single-crystal structure and packing arrangement of BISX. (c) Potential energy surface of BISX in the crystalline state and the optimized ground-state and excited-state geometries.”

2. Still on Figure 5, the authors only provide a qualitative picture of the computations. They need to provide the quantitative data, ie the S₁ and S₀ energies of the different computed minima.

Response: Thank you so much for your valuable comment. According to your suggestion, we have provided the experimentally quantitative data in Figure 5 and computationally quantitative data in Table S? In the solution state, the energy gap between S₁ and S₀ of CT_A and CT_{D-A} of BISX are 2.88 eV and 2.25 eV, respectively, which are obtained by the emission peak at low temperatures and very weak emission band at room temperature. In the aggregate state, the energy gap between S₁ and S₀ of CT_{D-A} is 2.61 eV obtained by the crystal emission. Thus, compared to the CT_{D-A} emission in solutions, CT_{D-A} in the aggregate state is closer to the Franck-Condon geometry. And the trend observed in the calculated data (Table S?) also agrees with the experimental results.

On page 16 of the revised manuscript,

Figure 5. (a) Potential energy surface of BISX in the THF solution and the optimized ground-state and excited-state geometries. (b) Single-crystal structure and packing arrangement of BISX. (c) Potential energy surface of BISX in the crystalline state and the optimized ground-state and excited-state geometries.”

On page 17 of the revised manuscript,

“And the obtained energy gap between S_1 and S_0 of CT_{D-A} (2.61 eV) is larger than that of in the solution state (2.25 eV), making CT_{D-A} in the aggregate state closer to the Franck-Condon geometry. And the trend observed in the calculated data (Table S4) also agrees with the experimental results.”

On page 33 of the revised SI,

Table S4. Theoretical calculations for fluorescence behaviors of BISX in the different states.

	$\lambda_{\text{computational}}$ (nm)	Energy gaps S_0-S_1 , computational (eV)
BISX _{THF solution, abs}	309.55	4.01
BISX _{THF solution, CTD-A}	397.79	3.12
BISX _{aggregates, abs}	310.21	4.00
BISX _{aggregates, CTD-A}	393.24	3.15

3. The authors argue that the CT_A and CT_{D-A} states differ in the probability of electron transition. This can be easily checked on the basis of the computed oscillator strengths. These data also need to be provided and discussed.

Response: Thank you so much for your valuable comment. We have provided the oscillator strengths in Table S3. It can be observed that the oscillator strength value of CT_{D-A} in solutions is as small as 0.0042, suggesting that CT_{D-A} in solutions is in a forbidden state. Though the conformation of CT_A cannot be obtained directly, an approximation can be made based on calculations of the isoquinolinone compound.

The oscillator strength of CT_A of isoquinolinone is 0.0159, which is significantly higher than that of CT_{D-A} (0.0042), indicating that electronic transitions in isoquinolinone compound mainly occur in the A part in the solution state.

On page 15 of the revised manuscript,

The calculated small oscillator strength value of 0.0042 also verified that (Table S3).

On page 33 of the revised SI,

Table S3. The oscillator strengths (*f*) of the S1 state of BISX and isoquinolinone compound.

	Oscillator strength (f)
BISX _{THF solution, abs}	0.0115
BISX _{THF solution, CTD-A}	0.0042
isoquinolinone _{THF solution, CTA}	0.0159

4. In the computational methods section, it is not clear how the structures in Figure S6 were obtained. The statement in page 21 states that single-point energy calculations were performed with M06-2X/6-31G(d), but not how the structures were optimized. Please add this information.

Response: Thank you so much for your valuable comment. We have supplemented the information.

On page 21 of the revised manuscript,

“Single-point energy calculations and all these structures were optimized using the M06-2X/6-31G (d)”

Minor point:

5. In page 15, in the statement: "While in the real case, such large torsion from 89.5° to 77.73° was hard to realize (...)", "real case" probably means "ground state", please correct.

Response: Thank you so much for your valuable comment. We have corrected it as below.

On page 15 of the revised manuscript,

“While in the ground state, such large torsion from 89.5° to 77.73° was hard to realize,”

Reviewers' Comments:

Reviewer #2:

Remarks to the Author:

The authors have addressed my comments satisfactorily, and I recommend publication.